# Maternal Stress, Early Life Factors and Infant Salivary Cortisol Levels

**DOI:** 10.3390/children9050623

**Published:** 2022-04-27

**Authors:** Caroline-Aleksi Olsson Mägi, Åshild Wik Despriee, Milada Cvancarova Småstuen, Catarina Almqvist, Fuad Bahram, Egil Bakkeheim, Anders Bjerg, Kari Glavin, Berit Granum, Guttorm Haugen, Gunilla Hedlin, Christine Monceyron Jonassen, Karin C. Lødrup Carlsen, Eva Maria Rehbinder, Leif-Bjarte Rolfsjord, Anne Cathrine Staff, Håvard Ove Skjerven, Riyas Vettukattil, Björn Nordlund, Cilla Söderhäll

**Affiliations:** 1Department of Women’s and Children’s Health, Karolinska Institutet, SE-171 77 Stockholm, Sweden; bjerg.anders@gmail.com (A.B.); gunilla.hedlin@ki.se (G.H.); bjorn.nordlund@ki.se (B.N.); cilla.soderhall@ki.se (C.S.); 2Astrid Lindgren Children’s Hospital, Karolinska University Hospital, SE-171 64 Stockholm, Sweden; catarina.almqvist@ki.se; 3Faculty of Medicine, Institute of Clinical Medicine, University of Oslo, NO-0424 Oslo, Norway; ashild.despriee@vid.no (Å.W.D.); g.n.haugen@medisin.uio.no (G.H.); k.c.l.carlsen@medisin.uio.no (K.C.L.C.); uxnnaf@ous-hf.no (A.C.S.); h.o.skjerven@medisin.uio.no (H.O.S.); riyas.vettukattil@medisin.uio.no (R.V.); 4Faculty of Health, VID Specialized University, NO-0424 Oslo, Norway; milasm@oslomet.no (M.C.S.); kari.glavin@vid.no (K.G.); 5Department of Medical Epidemiology and Biostatistics, Karolinska Institutet, SE-171 77 Stockholm, Sweden; 6Research Centre, Stockholm South General Hospital, SE-118 83 Stockholm, Sweden; fuad.bahram@ki.se; 7Division of Paediatric and Adolescent Medicine, Oslo University Hospital, NO-0424 Oslo, Norway; uxbake@ous-hf.no (E.B.); leif.bjarte.rolfsjord@sykehuset-innlandet.no (L.-B.R.); 8Martina Children’s Hospital, SE-114 86 Stockholm, Sweden; 9Department of Chemical Toxicology, Norwegian Institute of Public Health, NO-0213 Oslo, Norway; berit.granum@fhi.no; 10Division of Obstetrics and Gynaecology, Oslo University Hospital, NO-0424 Oslo, Norway; 11Faculty of Chemistry, Biotechnology and Food Science, Norwegian University of Life Sciences, NO-1430 Ås, Norway; christine.monceyron.jonassen@so-hf.no; 12Genetic Unit, Centre for Laboratory Medicine, Østfold Hospital Trust, NO-1714 Kalnes, Norway; 13Department of Dermatology and Vaenerology, Oslo University Hospital, NO-0424 Oslo, Norway; e.m.rehbinder@ous-research.no; 14Department of Paediatric and Adolescent Medicine Elverum, Innlandet Hospital Trust, NO-2381 Brumunddal, Norway

**Keywords:** salivary cortisol, perceived stress, infant, PreventADALL

## Abstract

Background: Salivary cortisol (SC), a commonly used biomarker for stress, may be disrupted by negative events in pregnancy, at birth and in infancy. We aimed to explore if maternal perceived stress (PSS) in or after pregnancy and SC levels in pregnancy were associated with SC in early infancy, and, secondly, to identify early life factors associated with infants’ SC levels (iSC). Methods: At 3 months of age, SC was analyzed in 1057 infants participating in a Nordic prospective mother-child birth cohort study. Maternal PSS was available from questionnaires at 18- and 34-week gestational age (GA) and 3-month post-partum, and SC was analyzed at 18-week GA. Early life factors included sociodemographic and infant feeding from questionnaires, and birth data from medical charts. Associations to iSC were analyzed by Spearman correlation and multinomial logistic regression analyses. Results: In this exploratory study neither PSS at any time point nor maternal SC (mSC) were associated with iSC. Higher birth weight was associated with higher levels of iSC, while inverse associations were observed in infants to a mother not living with a partner and mixed bottle/breastfeeding. Conclusions: Maternal stress was not associated with iSC levels, while birth weight, single motherhood and infant feeding may influence iSC levels.

## 1. Introduction

Salivary cortisol (SC) is a commonly used biomarker for stress in infants and children [1,2,3]. An association between maternal and infant saliva cortisol (iSC) levels has been observed in the early post-partum period and the infants’ first months in life [3,4], independent of sampling time of the day [5]. In adults and children, SC levels peak to their highest level after awakening, reflecting the cortisol awakening response (CAR) [6]. In infancy, CAR is not yet fully presented. The suggested maturation of CAR in infants is seen as a decrease in SC levels towards the evening, which usually occurs before 12 months of age [7].

In shared activities during the first year between mother and infant (e.g., breastfeeding and skin-to-skin care), SC levels can fluctuate in the infant and covary between the two [3,8]. Negative events in pregnancy can disrupt this covariation between mother and infant, leading to dyadic dysregulation and elevated iSC levels [9]. Psychological stressors to the infant have resulted in no to a small increase in iSC levels [10,11], whilst physical stressors have resulted in increased iSC levels [11]. In a case-control study of excessively crying infants, parents, but not their infants 0–5 months of age, had lower hair cortisol levels compared to controls [12], while the possible implication for infant stress of colic is not clear. We recently found that 26% of three months old infants from the general population-based Preventing Atopic Dermatitis and ALLergies in children (PreventADALL) cohort were perceived by their parents to have colic, abdominal pain or other discomforts [13], and this is thus a relatively common potential risk factor for stress. Fetal exposure to maternal stress in pregnancy (elevated plasma cortisol and psychosocial stress) may affect the regulation of infant stress and behavioral recovery after birth [14] and subsequently the maternal–infant bonding process [15]. Maternal perceived stress during pregnancy has also been suggested as a predictor of infant cortisol reactivity after birth and by 10 months of age [16]. In a recent study, we reported that 13–15% of women in the PreventADALL general population-based study experienced perceived high stress in mid- and late pregnancy [17]. However, possible associations between pregnancy-related factors, including maternal perceived stress and maternal saliva cortisol (mSC), birth-related factors, maternal stress post-partum or infant factors and early iSC levels in the general population are not well elucidated.

The aims of this exploratory study were to analyze the associations between maternal perceived stress during pregnancy and postpartum as well as pregnancy mSC levels and iSC levels at three months of age, and secondly to identify early life factors in pregnancy, at birth and in infancy that are associated with iSC levels at three months of age.

## 2. Materials and Methods

The women and infants in the present study are participants in the multicenter interventional birth cohort study PreventADALL (Preventing Atopic Dermatitis and ALLergy in children), conducted at the Oslo University Hospital and the Østfold Hospital Trust, Oslo, Norway; and Karolinska University Hospital, Stockholm, Sweden [18]. The PreventADALL study has two main objectives: to determine if allergic diseases may be prevented by skin and/or food interventions in infancy; and to identify factors early in life involved in non-communicable disease development. The present exploratory study adheres to the latter objective. Briefly, non-selected pregnant women from the general population were recruited antenatally between December 2014 and November 2016 at the routine ultrasound pregnancy screening at 18 weeks in mid-trimester (16–22 weeks gestational age), providing singleton or twin pregnancies and sufficient language skills (Norwegian or Swedish). Exclusion criteria were planning to move from the study area, severe maternal disease and pregnancies with three or more fetuses. Their infants were included at birth. Exclusion criteria also included severe neonatal disease and infant born before gestational week 35. Informed consent forms were signed by the women at enrolment and by both parents at birth. PreventADALL was registered at Clinical Trial Registration: ClinicalTrials.gov number: NCT02449850. Ethical approval was granted by the Regional Committee for Medical and Health Research Ethics in Norway, 8 December 2014 (2014/518) and the Swedish Ethical Review Authority, Sweden, 25 March 2015 (former Regional Ethics committee in Stockholm) (2014/2242-31/4).

### 2.1. Data Collection

At enrolment, a short interview was performed, height, weight and blood pressure were measured, a blood sample was drawn and a home saliva sampling kit was provided to the pregnant women. Electronic questionnaires were distributed twice in pregnancy; first at enrolment and the second around pregnancy week 34, based largely on questionnaires used in previous birth cohort studies [19,20,21]. The newborn infants of the participating mothers were included within the first 24 h after birth, with renewed informed consent by both parents. Electronic questionnaires at three months of age of the infant were completed by the mother. All completed questionnaires were submitted electronically and data stored at the University of Oslo, University Centre for Information Technology (USIT), Oslo, Norway.

From the 2697 pregnant women and their 2397 newborns enrolled in the PreventADALL study, three withdrew their consent, leaving 2394 mother-child pairs, whereof 2131 infants attended the three-month clinical examination. In the present study we included all 1057 infants who had available iSC measurement from saliva samples collected at three months of age (Figure 1).

### 2.2. Saliva Sampling and Cortisol Measurements

Home sampling kits were distributed at inclusion in pregnancy for maternal saliva sampling and at the three-month follow-up visit for infant saliva sampling. All kits included sampling instructions, a zip-locked plastic bag, a saliva kit (Salimetrics^®^, Carlsbad, CA, USA), a form to fill out the sampling time and date and prepaid addressed envelope to the study center (in Oslo for the Norwegian participants and in Stockholm for the Swedish participants). Parents were instructed to collect saliva within the next days, preferably on Monday, Tuesday or Wednesday, shortly after their own or the infant’s awakening (after 06.00) and before food/breastfeeding or water intake. The sample was refrigerated and mailed to the study center as soon as possible. Cortisol concentrations have previously shown stability in the postal service and to exposure to different temperatures [22,23]. Upon arrival, samples were registered and stored in −80 °C until analyzed.

Cortisol quantification was made with Radioimmunoassay (RIA), according to the manufacturer’s instructions, with CORT-CT2 kits from Cisbio Bioassays (Codolet, France) at Forskningscentrum (Södersjukhuset, Stockholm, Sweden) [24]. Salivary cortisol level was presented as nmol/L. The standard curve for SC measurement was 0.5–105 nmol/L. The standard procedure was to re-run all samples with a concentration higher than 110 nmol/L and dilute two-five times until concentrations were within the range of the standard curve. According to the manufacturer, the intra-assay precision of duplicates of saliva samples is CV (%) = 3.3 and inter-assay precision CV (%) = 8.4 (at Forskningscentrum, Södersjukhuset, Stockholm the intra-assay CV (%) = 5.13 and inter-assay CV (%) = 6.8).

Infant SC samples with a stated sampling time were collected between 02 AM and 9 PM (Appendix A). The main outcome for PSS and mSC correlation with iSC was iSC levels at three months of age given as nmol/L (regardless of sampling time), while quartiles of iSC (nmol/L) were used as the main outcome to explore early life factors in the same infants. SC were categorised for sensitivity analyses by sampling time; morning sampling (from 05.00–10.59); other sampling time (11.00–04.59) or missing sampling time.

### 2.3. Perceived Stress

Information on maternal stress was included in the electronic questionnaires and was measured with Cohen’s Perceived Stress Scale (PSS) (14 items), which scores from 0 to 4 for each of the 14 items with a total maximum score of 56 [25], and measures how often in the past month one’s life was “unpredictable, uncontrollable or overloaded” [25]. PSS has been validated in the Swedish language [26]. A higher score indicates higher perceived stress. With no universally accepted cut-off value for high stress, we used the value identified in a previous report from the PreventADALL study with high perceived stress defined by a score of 29 or higher [17].

### 2.4. Possible Factors Associated with Infant Saliva Cortisol

Early life factors collected through the electronic questionnaire in mid-pregnancy, included socio-demographic information, previous and current health and maternal stress, while information about the father and socio-demographic, health or disease changes in the mother since inclusion was captured in the 34-week GA questionnaire. The information included mothers not living with a partner; e.g., women who lived alone, divorced, widow, other, low maternal education (maximum 9/10-year school or high school), maternal age at inclusion and if it was the first pregnancy or the mother had previous deliveries (alive or still-born).

Birth-related factors were collected from the maternal and infant birth records and by PreventADALL study personnel at the inclusion visit of the infant. Induced labor was defined as a medically induced labor. Vaginal birth included unassisted and births assisted birth with forceps or vacuum extraction. Cesarean section included planned elective and acute cesarean section. Birth weight was the infant’s first recorded weight.

Infant factors included exclusively breastfed and mixed feeding by bottle and breastfeeding (mixed fed) as well as colic, abdominal pain and pain or other discomforts reported at three months of age based on the following question: “In the last three months, did the infant have any of following?” [13]. Furthermore, as participation in a randomized clinical trial with skin, and/or food interventions may be cumbersome, we included group allocation to one of four groups to which infants were randomized at birth; skin intervention (regular skin emollients by oil baths and facial cream at least five times per week from two weeks of age), food intervention (introduction of four allergenic foods from three months of age, alas not started at the time of data collection), skin and food intervention or no intervention, described in detail elsewhere [18,27]. Mothers who reported any smoking or use of snus (moist snuff) after birth were categorized as using nicotine.

### 2.5. Statistical Analysis

Infant SC levels included some outliers (data were positively skewed). Thus, descriptive statistics are presented with median, interquartile range (IQR) or minimum-maximum (min-max) (continuous variables) and counts and percentages (%) (categorical variables). Spearman non-parametric correlations were used to describe correlations between infant SC levels and maternal PSS and SC levels and are presented with Spearman’s coefficient rho (r) and *p*-value.

To explore early life factors that may be associated with iSC levels (regardless of sampling time or missing sampling time), multinomial logistic regression was used. Infant SC data were divided in four equally large groups ≤5.50 nmol/L (*n* = 266), 5.51–15.35 nmol/L (*n* = 263), 15.36–24.93 nmol/L (*n* = 265) and ≥24.94 nmol/L (*n* = 263). In the univariate and multivariate multinomial logistic regression models, the group with the lowest iSC levels (≤5.50 nmol/L) was used as the reference category. Variables with *p*-value less than 0.1 from univariate analyses (Appendix A) were included in the multivariate multinomial regression models as covariates, and the results are presented as odds ratios (OR) with 95% confidence intervals (95% CI) and *p*-value. Significance for the level in multivariate regression models was set at 0.05. All analyses were considered exploratory, thus no correction for multiple testing was performed.

In the Appendix A, Mann–Whitney U-test of factors possibly associated with iSC, using the entire study sample (Appendix A) as well as stratified by boys and girls (Appendix A), are presented as number, percent (%), median, inter-quartile range (IQR). T-test of included and not included infants is presented in Appendix A. Analyses were performed using SPSS statistics version 26 (IBM, Chicago, IL, USA).

## 3. Results

The 1057 included infants were similar to infants without available iSC samples (GA, age at saliva sampling, weight and maternal PSS) at three months of age and had a median age of 14 weeks and 51.4% were boys (Table 1). The overall median SC level was 15.30 nmol/L (IQR 15.35) (Table 1). Stratified by time of sampling, the median iSC levels were 14.00 nmol/L (IQR 13.40) among morning samples (*n* = 551), 13.20 nmol/L (IQR 16.1) in the 79 with other sampling time and 17.70 nmol/L (IQR 18.30) in the 427 infants with missing sampling time (Figure 2).

Maternal PSS was not significantly associated with iSC, neither by PSS in mid-trimester, third trimester or postnatally, nor among all infants (main outcome) or by sampling time (Table 2).

Maternal SC levels in mid-pregnancy was not significantly associated with iSC levels among all infants, nor among infants with morning sampling (Table 2). Further, mSC levels in mid-pregnancy were not associated with maternal PSS in mid-pregnancy, last trimester or postnatally (Appendix A).

Associations between early life factors and iSC levels are shown for univariate multinomial regression analyses in Appendix A. Mothers not living with a partner, PSS ≥ 29 reported at 34 weeks, cesarean section, birth weight and mixed-fed infants were included in the multivariate multinomial logistic regression models, as *p*-values were ≤0.1, while not living with a partner, birth weight and mixed-fed infants remained statistically significantly associated with iSC in the final model (Table 3). Infants of mothers not living with a partner had lower odds of iSC levels ≥ 15.4 nmol/L than infants of mothers who were cohabitant or married. Higher birth weight was significantly associated with an iSC of ≥ 24.9 nmol/L, while mixed-fed infants were less likely to have high iSC levels at three months of age (Table 3).

## 4. Discussion

In this exploratory study with 1057 infants from the general population, perceived stress during pregnancy and postpartum was not associated with iSC levels at three months of age. Infants of mothers living without a partner had a lower odd of higher iSC levels, as did infants fed with a combination of bottle as well as breastfeeding, while higher birth weight increased the odds of higher iSC levels at three months of age.

### 4.1. Early Life Factors and Infant Saliva Cortisol

The lack of association between maternal perceived stress during pregnancy or post-partum and iSC levels is an unexpected finding and has, to our knowledge, previously not been seen in other studies. The infant cortisol stress reactivity score has previously been associated with maternal stress [16] and maternal distress in pregnancy [28]. In contrast to our results, Leung et al. [16] found no association between iSC and feeding, health indicators or sociodemographic factors. We did not measure stress reactivity in the infant in this study. This could be a possible explanation as to why our results differ.

Salivary cortisol in the mothers in mid-pregnancy was not significantly associated with iSC. In contrast to our results, previous studies have shown associations between mSC levels in pregnancy or postpartum and their iSC levels from six months of age [2,3]. In a study by Nazzari et al. [29], higher levels of maternal pregnancy cortisol were associated with increased cortisol response in infants with mothers less emotionally available. Whereas Irwin [30] showed that elevated maternal cortisol in pregnancy was associated with infant cortisol reactivity at 6- and 12 months of age. Our results suggest that the possibility of identifying an association could be dependent on the timespan between maternal/infant samples, as well as the number of samples taken at each timepoint. We did not measure maternal caregiving nor cortisol reactivity to stress in the infant in our study, this may partly explain why no association was detected in mSC and iSC. Altogether, our findings suggest that the relationship between PSS and SC levels in mothers and iSC levels is complex, and that potential stressors and the timing of sampling is sensitive and standardization is lacking.

The lower iSC levels among infants with a mother not living with a partner compared to infants with both parents at home, is, to our knowledge, a novel finding. In adolescents, parental depression has been associated with a diurnal SC pattern [31], but children of single parents, chronically ill parents or healthy parents have not shown any difference in diurnal SC pattern [31]. In other settings, mothers not living with a partner could indicate low socioeconomic status. The Family Life project [32] follows the development of children in low-income families in poor rural counties in North Carolina and Pennsylvania. Longitudinal data on infant between-person and within-person SC from seven months until two years of age showed that higher levels of resting iSC collected during the day were associated with lower infant attention [33]. In the PreventADALL study the majority of participants are Scandinavian and have a higher socioeconomic status than The Family Life project participants [32], which makes the two studies difficult to compare.

We observed that an increased birth weight was associated with higher levels of iSC at three months. There is, to our knowledge, no previous data on this association. Previous findings on infant birth weight and mSC morning and evening levels or depressive symptoms have been shown in Lundholm et al. [34]. They found an association between birth weight and symptoms of maternal depression in the mother in early pregnancy in the Swedish MAESTRO cohort. In the American MADRES [35] cohort they did not see any significant association between infant birth weight and maternal third trimester cortisol levels.

At three months of age, there was no significant difference in median iSC levels between boys and girls, which is in accordance with the study by Rolfsjord et al. [1], in six months-old infants. However, they found higher morning iSC values in girls at two years of age. In other studies, the results are conflicting [36], although a recent review [36] showed higher cortisol levels and a steeper decline over the day in cortisol among girls compared to boys in the majority of the studies. The same study also suggested that girls express less fluctuation than boys over the day [36].

In our study, mixed-fed infants had lower iSC levels than exclusively breastfed infants. The benefits of breastfeeding in both mothers and their infants are well documented and explored. Mothers exclusively breastfeeding their infants have shown lower SC levels postpartum than mothers with mixed-fed infants [37]. In our study, mothers not living with a partner did not differ from cohabitant or married mothers in breastfeeding status (results not presented). The suggested fluctuation in iSC levels during breastfeeding [3] could explain why we found lower iSC levels among the mixed-fed infants. In addition, mothers who breastfeed report less anxiety, negative mood, stress and show reduced SC levels compared to mothers with formula-fed infants [38]. Breastfeeding overall is high in the PreventADALL study, where 93.6% of the women breastfed their infant to some extent by three months [39]. The latter could possibly explain why we did not detect a correlation, as previously described by Jonas et al. [3], between SC levels in mothers and infants, but lower iSC levels in the mixed-fed infants. Jonas et al. [3] showed a positive correlation of iSC levels in breastfed infants and mSC levels during pregnancy and post-partum [3]. Their sampling method differs from ours, with saliva samples taken at several occasions before and after breastfeeding, which could further explain why no correlation was found with our sampling method.

At the three-month clinical visit in the PreventADALL study, many of the parents randomized to oil baths mentioned the bathing event as a positive experience. There was no significant difference in iSC levels between the intervention groups, but the lowest iSC values were found among boys in the intervention group of oil baths. A previous study has found that bathing the infant promotes parent–infant bonding [40], hence closeness to the mother and/or father. To capture the possible positive effect of this bonding in a bathing situation, a different, repeated iSC sampling procedure closer in time before and after the bath may be needed. Other approaches, e.g., qualitative methods, could also contribute to the knowledge on parent–infant bonding in bathing situations.

Infants with colic and abdominal pain showed no difference in iSC levels from infants without colic or abdominal pain in our study. Mostly, our participants’ saliva was sampled during the day, while colic most commonly debuts in the early hours of the night. White et al. [41] have previously described a blunted cortisol rhythm in infants with colic, but their overall cortisol levels in infants with colic did not differ from their controls. Although physical pain has been a suggested stressor in infants [11], colic and abdominal pain is a common stressful concern for parents [42]. Infants with abdominal pains and colic could perhaps affect maternal PSS.

### 4.2. Strengths and Limitations

The present study has several strengths. All samples were obtained in the home environment by a parent or guardian, providing a feasible way to collect saliva from a large number of infants, in contrast to collection e.g., plasma cortisol that requires a more comprehensive procedure. Furthermore, saliva sampling is unlikely to cause stress to the mother or infant [43]. All the iSC and mSC analyses were performed by trained personnel with long experience of this particular SC analysis from other cohorts (Forskningscentrum, Södersjukhuset, Stockholm, Sweden). Although the questions in the electronic questionnaires are not validated, the majority of questions are standard questions used in other birth cohort studies [19]. Furthermore, we followed our participants (mothers, mother–child pairs) from early pregnancy and onward. Our study included longitudinal data from pregnancy, at birth and by three months infant age. The obstetric outcomes were verified by reviewing the individual hospital medical charts (Oslo and Kalnes) and The Swedish Pregnancy Register.

The following limitations should be noted. The population in this exploratory study originated from the PreventADALL birth cohort, where the main objective focuses on allergic disease prevention and the development of non-communicable diseases. The proportion of participants with a heredity of allergic diseases could therefore be biased. Furthermore, we used one validated instrument to measure stress in the mothers. That makes the generalizability of this study limited. The sampling method used resulted in a large amount of missing saliva samples (1057/2131) and missing records of sampling time of the infant saliva due to the home-sampling procedure. All iSC were analysed without any consideration of sampling time and skewness. SC samples were not tested for blood contamination. Our participants were instructed to collect the saliva sampling in the morning, 30 min after the child’s or own awakening, preferably before breast- or bottle-feeding. We did not collect complementary information nor further investigate if the instructions to study protocol were followed. Our stress data were based on one validated instrument and it was collected through self-reported electronic questionnaires which could increase the risk of information bias. PSS has been translated but not validated into the Norwegian language. Several samples in our study were taken later in the morning, in the afternoon or had a missing sampling time. We did not see, nor could draw conclusions of, a cortisol circadian rhythm in our infant population, neither on a group level nor on an individual level as our study only generated one sample per infant. The intention was to include a general population in this study; however, socioeconomic status was higher among the infants’ families who provided a saliva sample than those who did not (not presented), and data on maternal depression- and anxiety-medication were not available for the mothers. The association between iSC in non-cohabitant mothers was based on a very limited number of individuals, thus might be sporous. Limited statistical power did not allow us to estimate these associations with sufficient precision. All analyses in this study were considered exploratory so no correction for multiple testing was performed. These findings are therefore explorative and need to be confirmed in other studies with similar sampling and settings.

Our results suggest that the possibility of identifying a correlation between PSS and iSC could be dependent on the timespan between maternal/infant samples as well as the number of samples taken at each timepoint.

## 5. Conclusions

In this exploratory study, maternal perceived stress in pregnancy and post-partum was not associated with iSC levels at three months of age. Early life factors which were significantly associated with iSC included increased birthweight with increased iSC levels at three months, while infants of mothers not living with a partner and mixed-fed infants had lower levels of iSC. Altogether, these results suggest that the relationship between PSS and mSC levels in mothers and early life factors of iSC levels in infants are complex.

## Figures and Tables

**Figure 1 children-09-00623-f001:**
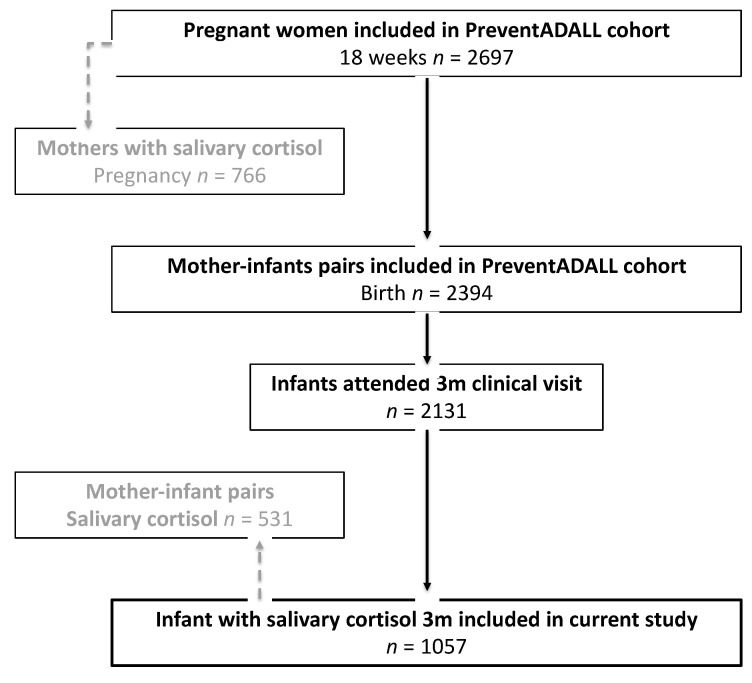
Flowchart of included mothers and infants.

**Figure 2 children-09-00623-f002:**
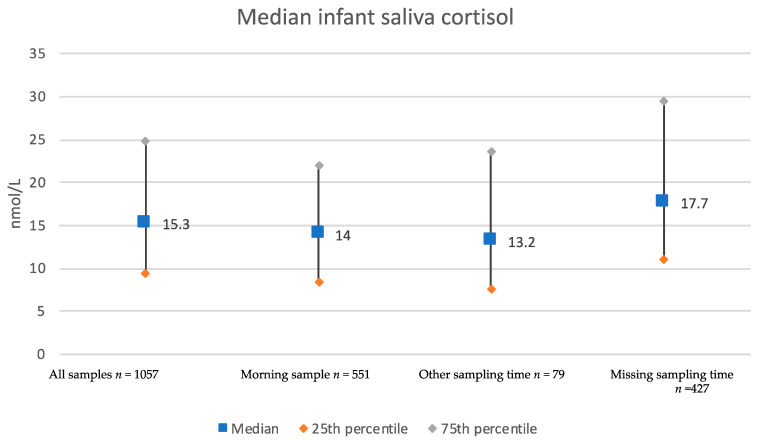
Infant saliva cortisol levels related to sampling time (*n* = 1057). Values are given in median (nmol/L), blue large squares) and inter quartile range (25th—75th percentile).

**Table 1 children-09-00623-t001:** Maternal, neonatal and infant characteristics of the infants with saliva cortisol at three months of age (*n* = 1057) and for those that did not provide saliva samples (not included *n* = 1337). Values are presented as numbers, percent (%), median iSC, interquartile range (IQR).

		Study Cohort (*n* = 1057)		Not Included (*n* = 1337)	
		*n* (%)	Median (IQR)	*n* (%)	Median (IQR)
Gestational age, birth (weeks)		1052 (99.5)	40.14 (1.71)	1336 (99.9)	40.29 (1.71)
Birth weight (grams)		953 (90.2)	3550 (646)	1332 (99.6)	3555 (620)
Age at SC sampling (weeks)		663 (62.7)	14.00 (2.0)	674 (50.4)	14.00 (4.0)
Saliva cortisol, infant (nmol/L)		1057 (100)	15.30 (15.35)	1337(100)	Na *
Boys		543 (51.4)	15.50 (15.9)	712 (53.3)	Na *
Girls		514 (48.6)	15.10 (14.58)	625 (46.7)	Na *
Maternal age (years)		1057 (100)	32 (6.00)	1337(100)	32 (6.00)
Saliva cortisol, maternal 18 w (nmol/L)		531 (50.2)	29.60 (14.30)	190 (14.2)	29.4 (13.35)
Perceived stress scale					
Pregnancy, 18 weeks		1002 (94.7)	21.00 (9.00)	1169 (87.4)	21.00 (10.00)
Pregnancy, 34 weeks		1016 (96.1)	20.00 (10.00)	1219 (91.2)	20.00 (9.00)
3-month post-partum		933 (88.2)	18.00 (10.00)	919 (68.7)	18.00 (9.00)
		*n* (%)	median (IQR)	*n* (%)	median (IQR)
Marital status, *n* (%)	Married	430 (42.9)	15.35 (16.0)	465 (39.8)	Na *
Cohabitant		549 (54.8)	15.80 (15.7)	668 (57.1)	Na *
Not living with a partner		23 (2.3)	7.70 (10.2)	36 (3.1))	Na *
Education *n* (%)	High school	106 (10.6)	17.50 (17.25)	133 (11.5)	Na *
≤4 years University		318 (31.8)	15.75 (16.4)	133 (32.0)	Na *
>4 years University, PhD		575 (57.6))	14.90 (13.9)	656 (56.5)	Na *

* not applicable.

**Table 2 children-09-00623-t002:** Spearman non-parametric correlations between infant cortisol at three months and maternal perceived stress in pregnancy at 18-, 34 weeks, and at three months infant age and maternal cortisol (18 w). Groups represent infants with All infant saliva cortisol (All iSC) regardless of sampling time (*n* = 1057); *Morning sample* (05–10:59, *n* = 551); *Other sampling time* from 11 am (until 04:59 pm, *n* = 79); *Missing sampling time* (no stated time of sampling, *n* = 427).

Infant Saliva Cortisol Samples								
	All		Morning Sample		Other Sampling Time		Missing Sampling Time	
	*n*	r (*p*-Value)	*n*	r (*p*-Value)	*n*	r (*p*-Value)	*n*	r (*p*-Value)
PSS 18 weeks	1002	0.00 (0.99)	516	−0.07 (0.10)	70	−0.06 (0.65)	416	0.06 (0.26)
PSS 34 weeks	1016	−0.01 (0.66)	533	−0.06 (0.14)	76	0.00 (0.96)	407	0.05 (0.28)
PSS 3-month	933	−0.04 (0.25)	496	−0.07 (0.13)	67	−0.03 (0.83)	370	−0.00 (0.98)
Maternal cortisol 18 w	531	0.05 (0.29)	369	0.07 (0.18)	51	0.17 (0.23)	111	0.07 (0.47)

**Table 3 children-09-00623-t003:** Multivariate multinomial logistic regression model analysis of possible associations of early life factors (identified in univariate multinomial regression, *p*-value < 0.1) and salivary cortisol in 1057 infants at three months of age.

Multivariate Analysis <0.1 from Univariate (Reference Category <5.50 nmol/L, *n* = 266)									
	Saliva Cortisol 5.51–15.35 nmol/L (*n* = 263)			Saliva Cortisol 15.36–24.93 nmol/L (*n* = 265)			Saliva Cortisol ≥ 24.94 nmol/L (*n* = 263)		
	OR	95% CI	*p*	OR	95% CI	*p*	OR	95% CI	*p*
Birth weight *	1.04	1.00–1.08	0.09	1.03	0.99–1.07	0.12	1.05	1.01–1.09	0.02
Cesarean section	0.64	0.37–1.09	0.10	0.97	0.59–1.59	0.91	1.11	0.68–1.81	0.67
Vaginal birth (ref.)	1.00			1.00			1.00		
Not living with partner	0.48	0.23–1.01	0.05	0.32	0.14–0.73	0.01	0.46	0.22–0.98	0.04
Married/cohabitant (ref.)	1.00			1.00			1.00		
Mixed fed	1.02	0.69–1.50	0.93	0.72	0.50–1.06	0.09	0.60	0.41–0.88	0.01
Breastfed only (ref.)	1.00			1.00			1.00		
High PSS 34 w	0.98	0.55–1.72	0.93	1.43	0.84–2.45	0.19	0.81	0.44–1.47	0.48
Low stress < 29 (ref.)	1.00			1.00			1.00		

* Represents increase in 100 g.

## Data Availability

Participants of this study were not asked to consent for open access data from third parties.

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
