# Peer review of "Maternal Stress, Early Life Factors and Infant Salivary Cortisol Levels"

_children, 2022, doi:10.3390/children9050623_

Round 1

Reviewer 1 Report

An addition to an already substantial literature. Good to use pregnancy cohort data. Intention to elucidate possible associations commendable.

Manuscript requires editing for minor grammatical mistakes and typographical errors.

Scant background literature review to put current study in context. No mention of previous findings of no relationship between maternal stress, cortisol, and infant outcomes.

Saliva samples undertaken by participants, refrigerated and then mailed to study centre – potential for spoiling? Reference(s) in support of this practice?

Reliability and validity of cortisol analysis as this varies by method used? Were women using anti-anxiety or anti-depressant medication or taking other medications that might interfere with their experience of stress or cortisol metabolism? Given that the circadian rhythm of cortisol is not established until after 12 months, was it wise to use SC measures taken at different times during the day. Do the measures taken accurately reflect those of previous studies?

Maternal stress ascertained using one validated measure. However, in studies of this type it is usual to use multiple instruments to measure state and trait anxiety, depression and life stress in addition to perceived stress. Why use a cut-off for the PSS rather than treat it as a continuous variable?

Other than basic socio-demographic information it is usual, and it would have been very helpful, to include maternal measures of financial strain, relationship quality and social support. BMI is another factor with a known association with cortisol.

Was there missing data? How was this managed? Imputation?

Why was cortisol made a categorical variable, when it is clearly continuous and why some form of linear regression wasn’t used with log SC. Why use logistic regression? The analysis that was undertaken is at the least, very basic. It is not the complex analysis that would be required to “elucidate possible associations”.

In keeping the discussion very brief, the current findings have not been placed in appropriate context. Previous findings have been referred to only when they support the argument being made. It would be much better to undertake and integrative review and then compare and contrast findings with all relevant literature.

The study has many significant limitations as identified by the authors.

Author Response

An addition to an already substantial literature. Good to use pregnancy cohort data. Intention to elucidate possible associations commendable.

  1. Manuscript requires editing for minor grammatical mistakes and typographical errors.

Response: Thank you, we have had an external native English speaker read and revise the manuscript.

  1. Scant background literature review to put current study in context. No mention of previous findings of no relationship between maternal stress, cortisol, and infant outcomes.

Response: To our knowledge, infant saliva cortisol and maternal perceived stress has previously not been explored in a larger cohort similar to ours. We have added a sentence on maternal perceived stress and infant saliva cortisol to the background. Line: 64-66

“Maternal perceived stress during pregnancy has also been suggested as a predictor of infant cortisol reactivity after birth and by 10 months age (16).”

Leung E, Tasker SL, Atkinson L, Vaillancourt T, Schulkin J, Schmidt LA. Perceived maternal stress during pregnancy and its relation to infant stress reactivity at 2 days and 10 months of postnatal life. Clin Pediatr (Phila). 2010;49(2):158-65.

  1. Saliva samples undertaken by participants, refrigerated and then mailed to study centre – potential for spoiling? Reference(s) in support of this practice?

Response: Thank you for this valuable comment. Saliva cortisol is commonly taken in the home environment and mailed trough postal service or brought to study site by the participants. Cortisol is considered stable in room temperature for at least one week. In a previous study, participant saliva was tested for cortisol concentrations after conditions similar to postal service. They found that cortisol concentrations were stable during extended periods without freezing when exposed to varying temperatures and movement (Clements AD, Parker CR. The relationship between salivary cortisol concentrations in frozen versus mailed samples. Psychoneuroendocrinology 1998;23:613–6).

Furthermore, another study has concluded that thawing/freezing did not appear to affect the concentrations of salivary cortisol and that it may be stored for at least a year at -20°C and -80°C. Usage of salivette tampons and storage of centrifuged samples has previously been evaluated by Garde, A. H. and A. M. Hansen (2005). "Long-term stability of salivary cortisol." Scand J Clin Lab Invest 65(5): 433-436.). We have added a sentence regarding this in the methods section, based on the two studies above. Lines: 125-126

“Cortisol concentrations has previously shown stability in conditions similar to postal service and exposure to different temperatures (22, 23).”

  1. Clements AD, Parker CR. The relationship between salivary cortisol concentrations in frozen versus mailedsamples. Psychoneuroendocrinology 1998;23:613–6
  2. Garde, A. H. and A. M. Hansen (2005). "Long-term stability of salivary cortisol." Scand J Clin Lab Invest 65(5): 433- 436.
  3. Reliability and validity of cortisol analysis as this varies by method used?

Response: Thank you for pointing this out as it is a relevant question. We did not test the cortisol analysis reliability and validity in our sample. In our study, saliva cortisol levels were measured with RIA (radioimmunoassay) (Kirschbaum C, et al. 1989), which is a commonly used for saliva cortisol measurements. When diagnosing Cushing’s syndrome, saliva cortisol measured by RIA gave results closer to the expected value than EIA (EIA overestimated the concentrations) (Raff, H et al 2002). Our samples were analysed at Forskningcentrum at Södersjukhuset (Stockholm, Sweden). They have previously performed the same analyses for several other cohorts (other than ours, references not presented). In Hansen. Å, M. et al. radioimmunoassay for salivary cortisol measurements was evaluated in 120 healthy individuals with reference material in water. The effect of age, BMI, diurnal variation, gender, days of sick leave and smoking habits was established. The method evaluation of reference (water) recovery was 97% (did not show any bias of the method). The within subject variation was 0.14 (adequate for healthy subjects) and salivary cortisol was not affected by either of the effects tested. This reference has been added to the manuscript. Line 131.

References:

(Kirschbaum C, Strasburger CJ, Jammers W, Hellhammer DH. Cortisol and behavior: 1. Adaptation of a radioimmunoassay kit for reliable and inexpensive salivary cortisol determination. Pharmacol Biochem Behav. 1989 Dec;34(4):747-51. doi: 10.1016/0091-3057(89)90269-4. PMID: 2623029.)

(Raff, H., et al. (2002). "Comparison of two methods for measuring salivary cortisol [7]." Clinical chemistry 48: 207-208.)

 (Evaluation of a radioimmunoassay and establishment of a reference interval for salivary cortisol in healthy subjects in Denmark. / Hansen, Åse Marie; Garde, A H; Christensen, J M; Eller, N H; Netterstrøm, B. I: Scandinavian Journal of Clinical & Laboratory Investigation, Bind 63, Nr. 4, 2003, s. 303-10.)

  1. Were women using anti-anxiety or anti-depressant medication or taking other medications that might interfere with their experience of stress or cortisol metabolism?

Response: This PreventADALL cohort mainly focuses on allergic diseases, hence we did not ask specifically about anti-anxiety or anti-depressant medication. The mothers did however report medications in free text at inclusion. Among the mothers with available saliva sample: 5/722 took antidepressant and 2 took medications primary intended for other conditions but could also have been used for depression. These numbers are small (<1%) and unlikely to affect the results. 28/722 mothers reported Levaxin use that could interfere with cortisol. However, when prescribing Levaxin, controlling for and monitoring cortisol levels is crucial since the medication can lower cortisol. These numbers have not been added to the manuscript as these data are unpublished and has not yet been cleaned. However, we have added it to the limitations of the study. Line: 364-365

“…and data on maternal depression- and anxiety-medication for the mothers were not available.”

  1. Given that the circadian rhythm of cortisol is not established until after 12 months, was it wise to use SC measures taken at different times during the day. Do the measures taken accurately reflect those of previous studies?

Response: To our knowledge, no studies on single samples of saliva cortisol in infancy without e.g. predefined setting with physical or psychological stressor to the infant or infant-mother dyad has been done. Our study median of 14 nmol/L (morning sample) and 15.3 nmol/L (all 1057 infants) is higher than e.g. Ivars (Ivars et al. 2015) morning median SC was 7.5 nmol/L at three-months and 8 nmol/l at four months. In Rolfsjord et al. (2017) when establishing reference values, median morning SC levels was 23.7nmol/L (infants age 0-13 months, n=199). Since we did not find any correlation between maternal perceived stress or maternal saliva cortisol levels and infant cortisol levels in the different time groups, we did not stratify for time in our analyses.

The participants in our study did not always follow the study protocol. This is as stated in the manuscript, a limitation of our study. However, as the circadian rhythm is not established yet by the age of three months, this made it possible for us to use all collected saliva samples in our analysis without stratifying for time or exclude saliva samples that were collected later in the day, or at an unknown timepoint.

Reference:

(Ivars, K., et al. (2015). "Development of Salivary Cortisol Circadian Rhythm and Reference Intervals in Full-Term Infants." Plos One 10(6): e0129502.)

(Rolfsjord, L. B., et al. (2017). "Morning Salivary Cortisol in Young Children: Reference Values and the Effects of Age, Sex, and Acute Bronchiolitis." The Journal of Pediatrics 184(C): 193-198.e193.)

  1. Maternal stress ascertained using one validated measure. However, in studies of this type it is usual to use multiple instruments to measure state and trait anxiety, depression and life stress in addition to perceived stress. Why use a cut-off for the PSS rather than treat it as a continuous variable?

Response: Thank you for this valuable observation. This study originates from The PreventADALL birth cohort. The two main objectives of PreventADALL were to determine if allergic diseases may be prevented by skin and/or food interventions in infancy, and to identify factors early in life involved in non-communicable disease development. We agree that it would have been valuable to have multiple instruments to measure depression, anxiety or life stress in the mothers. PreventADALL is a longitudinal cohort, and questionnaires to be filled out at several timepoints. Therefore, it was unfortunately not realistic to add more than one validated measurement for stress.

In a previous study in the PreventADALL-cohort we studied allergic disease and stress in pregnancy (Olsson Magi, C. A., et al. (2020)).In that study we used a cut-off for high perceived stress (PSS). High PSS was defined as 1 SD above mean PSS at pregnancy week 18 (³29). In this study, we used both continuous and categorical PSS (low PSS or high PSS) in univariate multinomial regression analysis (Supplementary 2). Only high PSS had a p-value of less than 0.1 and were included in the multivariate multinomial regression of possible associations with infant saliva cortisol (Table 3).

Previous studies including high PSS:

(Olsson Magi, C. A., et al. (2020). "Allergic disease and risk of stress in pregnant women: a PreventADALL study." ERJ Open Res 6(4).).

  1. Other than basic socio-demographic information it is usual, and it would have been very helpful, to include maternal measures of financial strain, relationship quality and social support. BMI is another factor with a known association with cortisol.

Response: Thank you for this comment. Since the original study design in PreventADALL did not include questions about financial strain, relationship quality or social support, this is unfortunately not possible. We agree that for maternal perceived stress; socio-demographic factors as mentioned above are very important. In our previous study on maternal stress and allergic disease in pregnant women we saw e.g. lower income, changes in employment and moving to new residence were associated with higher stress in pregnant women (Olsson Magi, C. A., et al. (2020).

In our univariate multinomial regression in this study, we include factors that could potentially describe socio-demographics such as “mothers not living with a partner” vs “married/cohabitant mothers”, “maternal education” (high school only vs higher education), and “low yearly family income” (<600.000 NOK/SEK » 58.000 dollars). Our univariate analysis showed that only “mothers not living with a partner” had a univariate association (p-value < 0.1) to infant saliva cortisol levels at three-months age (Supplementary 2). And later also a p-value of 0.01 and 0.04 in the multivariate multinomial regression of possible associations with infant saliva cortisol levels (Table 3).

We included infant weight instead of infant BMI since infant BMI is not commonly used in the clinic in the pediatric population in infancy. Maternal BMI was not included in the univariate multinomial regression. Saliva cortisol levels in the adult population has previously been shown not to be affected by BMI (Hansen et al. 2003).

Reference:

(Olsson Magi, C. A., et al. (2020). "Allergic disease and risk of stress in pregnant women: a PreventADALL study." ERJ Open Res 6(4).).

(Evaluation of a radioimmunoassay and establishment of a reference interval for salivary cortisol in healthy subjects in Denmark. / Hansen, Åse Marie; Garde, A H; Christensen, J M; Eller, N H; Netterstrøm, B. I: Scandinavian Journal of Clinical & Laboratory Investigation, Bind 63, Nr. 4, 2003, s. 303-10.).

  1. Was there missing data? How was this managed? Imputation?

Response: Inclusion criteria in our study was infant with saliva cortisol, therefore, we did not include any infants with missing values on saliva cortisol. Thus, there were no missing values on the main outcome. However, some of the possible predictive factors and background variables  had a few missing values. Given the proportion of missing values was so limited, we did not make any imputation for the missing data. To assess representability, we have added a Supplementary Table (Supplementary Table 5.) with t-test between included infants (with SC) and not included (did not provide a SC sample) for transparency between the main PreventADALL cohort (n=2394) and the included infants with SC sample in this study (n=1057).

  1. Why was cortisol made a categorical variable, when it is clearly continuous and why some form of linear regression wasn’t used with log SC.

Response: Thank you for this observation. Similar to many biological variables, cortisol had indeed a very skewed distribution. We agree with the reviewer that logarithmic transform might normalize distributions of log-normally distributed variables, however this was not the case of cortisol variable in our sample. In addition, our aim was to identify possible association between selected covariates and having cortisol levels over a threshold indicating being stressed. Thus, in line with the aims, we categorized cortisol variable and treated it as a nominal outcome. Moreover, our data included outliers which were still within the plausible range of cortisol values but would make it difficult to analyze cortisol as a continuous variable even when log transformed. This could potentially increase the risk of over interpreting the results.

  1. Why use logistic regression? The analysis that was undertaken is at the least, very basic. It is not the complex analysis that would be required to “elucidate possible associations”.

Response: We agree with the reviewer that our data required a more complex analytical approach than using logistic regression. Therefore, we used multinomial logistic regression. This is clearly described in the Statistical analyses section (Line: 177-190). During preliminary analyses, when analyzing the data we did consider quantile regression and ordinal regression. However, multinomial regression was chosen because the outcome was a nominal variable and a multinomial regression fitted well to our data.

  1. In keeping the discussion very brief, the current findings have not been placed in appropriate context. Previous findings have been referred to only when they support the argument being made. It would be much better to undertake and integrative review and then compare and contrast findings with all relevant literature.

Response: Thank you for this comment. We do not fully agree with you, we found our results difficult to confirm in the previously scanned literature because e.g. different methods used, different outcomes than we used in our study and those we referred to.

Line 242-246

Salivary cortisol in the mothers in mid-pregnancy was not significantly associated with iSC. In contrast to our results previous studies have shown associations between mSC levels in pregnancy or postpartum and their iSC levels from six months age (2, 3).

Line: 275-276, 278-281

At three months of age, there was no significant difference in median iSC levels between boys and girls…. the results are conflicting (35), although a recent review (35) showed higher cortisol levels and a steeper decline over the day in cortisol among girls compared to boys in the majority of the studies. The same study also suggested that girls express less fluctuation than boys over the day (35).

We have added a sentence: Line: 257-259

“In contrast to our results, Leung et al. (16) found no association between iSC and feeding, health indicators, or sociodemographic.”

  1. The study has many significant limitations as identified by the authors.

Response: We agree with the reviewer.

Reviewer 2 Report

This is a very well written and well executed study.

I have noted some small changes for your consideration in relation to english grammar and use.

Author Response

This is a very well written and well executed study.

I have noted some small changes for your consideration in relation to english grammar and use.

Response: Thank you for this kind review. An external examiner has read and commented on the grammar. We have changed the text and wording after your suggestions.

Reviewer 3 Report

1. What is the main question addressed by the research?

The prediction of infant stress assessed by salivary cortisol (SC) based on maternal stress indicators, including maternal SC and percieved stress, and other infant and care factors.

2. Do you consider the topic original or relevant in the field, and if
so, why?

Infant stress is an important outcome and it's always interesting to better understand its determinants. The study is a secondary analysis of data collected within an allergy clinical trial of mother-infant dyads and comprises a large compelling dataset. 

3. What does it add to the subject area compared with other published
material?

The longituindal aspects and the biological measurements of SC for both mothers and infants makes this study noteworthy. However the null and counterintuitive findings from the multinomial log. regressions, esp. concerning the inverse associations between infant SC and non-cohabitation and mixed feeding, is difficult to explain.

4. What specific improvements could the authors consider regarding the
methodology?

lines 128-9: the primary outcome for the analyses appear to be the infant SC quartiles. Please clarify.

In general it might help readability if the terms infant SC or the abbrev. ISC and maternal SC or MSC were consistently used to differentiate these variables.  

Provide in the main text select descriptive statistics to help characterise the sample, esp. concerning the factors that are sig. in the final multinomial model. Some of this info is buried in one of the supplemental tables but is difficult to identify. 

Clarify why some of the predictive factors are entered in the scales that they were. Why was PSS entered as a categorical factor and not as a continuous one? The rationale for using categories for ISC was because of outliers, but please clarify in general the choice of scale of measurment used for other factors. 

Table 3: there's a marginal effect with not living with a partner that should be highlighted as well.

Was ordinal logistic regression considered as an analysis? 

5. Are the conclusions consistent with the evidence and arguments
presented and do they address the main question posed?

Overall yes, however:

lines 228-9, 248-9: novel finding may not be the best term here. The findings were null or in unexpected directions. As is typical in these situations, it's not at all clear why this was the case. Perhaps the authors are correct that there were cultural/social structural factors that have mitigated or reversed some associations, but it is not knowable here.  

6. Are the references appropriate?

Yes.

7. Please include any additional comments on the tables and figures.

Tables in general need better formatting for clarity- they're difficult to read across multiple pages. Sometimes the headings contain notes that should actually be elaborated in a formal table footnote for clarity. In other cases a table note should better explain what is being presented. For ex., Supplemental Table 3 presents sig. values for pairwise test against a reference catgeory within each factor. Suppl. 4 presents tests within boy and girl strata.

Author Response

  1. What is the main question addressed by the research?

The prediction of infant stress assessed by salivary cortisol (SC) based on maternal stress indicators, including maternal SC and perceived stress, and other infant and care factors.

  1. Do you consider the topic original or relevant in the field, and if
    so, why?

Infant stress is an important outcome and it's always interesting to better understand its determinants. The study is a secondary analysis of data collected within an allergy clinical trial of mother-infant dyads and comprises a large compelling dataset. 

  1. What does it add to the subject area compared with other published
    material?

The longitudinal aspects and the biological measurements of SC for both mothers and infants makes this study noteworthy. However, the null and counterintuitive findings from the multinomial log. regressions, esp. concerning the inverse associations between infant SC and non-cohabitation and mixed feeding, is difficult to explain.

Response: Thank you for this valuable comment. We fully agree with the reviewer. The association between iSC in non-cohabitant mothers was based on a very limited number of individuals thus we agree it can be sporous. We fully acknowledge that we had limited statistical power to estimate these associations with sufficient precision. In addition, we found it difficult to explain it have added the following to the limitations of the study. Line: 342-344

“The association between iSC in non-cohabitant mothers was based on a very limited number of individuals thus might be sporous. Limited statistical power did not allow us to estimate these association with sufficient precision.”

4.a. What specific improvements could the authors consider regarding the
methodology?

lines 128-9: the primary outcome for the analyses appear to be the infant SC quartiles. Please clarify.

Response: Thank you for pointing this out. We have now clarified this in the manuscript. Lines: 132-136

The main outcome for PSS and mSC correlation with iSC was iSC levels at three months of age given as nmol/l (regardless of sampling time), while quartiles of iSC (nmol/l) were used as main outcome to explore early life factors in the same infants. SC were categorised for sensitivity analyses by sampling time; morning sampling (from 05.00-10.59), other sampling time (11.00-04.59) or missing sampling time.

Previous text:The main outcome was SC levels at three months of age given as nmol/l (regardless of sampling time), while secondary outcomes were quartiles of SC (nmol/l) in the same infants. SC were categorised for sensitivity analyses by sampling time; morning sampling (from 05.00-10.59), other sampling time (11.00-04.59) or missing sampling time.”

4.b. In general it might help readability if the terms infant SC or the abbrev. ISC and maternal SC or MSC were consistently used to differentiate these variables.  

Response: We thank the reviewer for this suggestion. We have changed “infant SC” into “iSC” and “maternal SC” into “mSC” which we think improve readability and consistency.

4.c. Provide in the main text select descriptive statistics to help characterise the sample, esp. concerning the factors that are sig. in the final multinomial model. Some of this info is buried in one of the supplemental tables but is difficult to identify.

Response: Thank you for pointing this out. We have added descriptive statistics (marital status and maternal education in three categories each) in Table 1.

4.d. Clarify why some of the predictive factors are entered in the scales that they were. Why was PSS entered as a categorical factor and not as a continuous one? The rationale for using categories for ISC was because of outliers, but please clarify in general the choice of scale of measurement used for other factors.

Response: We have in previous studies (Olsson Mägi et a. 2020. Despriee et al 2021) used a cut-off for high stress (1 SD above mean) in our PreventADALL population. In this study, we used both continuous and categorical PSS (low or high) in univariate multinomial regression (Supplementary 2). Only high PSS had a p-value of less than 0.1 and were included in the multivariate multinomial logistic regression of possible associations with infant saliva cortisol (Table 3). Other factors such as yearly income, marital status, parity, tobacco use, birth method has previously been used as a dichotomous (yes/no) variable in studies from PreventADALL. Breastfeeding was common in PreventADALL 93.6 % breastfed to some extent (Nordhagen et al. 2020). Therefore, it was categorized as exclusive breastfed (or mixed fed) to elucidate exclusive breastfeeding.

Previous studies including high PSS, yearly income, marital status, colic, abdominal pains, breastfeeding:

Olsson Magi, C. A., et al. (2020). "Allergic disease and risk of stress in pregnant women: a PreventADALL study." ERJ Open Res 6(4).

Despriee, Å. W., et al. (2021). "Prevalence and perinatal risk factors of parentreported colic, abdominal pain and other pain or discomforts in infants until 3 months of age A prospective cohort study in PreventADALL." Journal of Clinical Nursing.

Nordhagen, L. S., et al. (2020). "Maternal use of nicotine products and breastfeeding 3 months postpartum." Acta Paediatr 109(12): 2594-2603.

4.e. Table 3: there's a marginal effect with not living with a partner that should be highlighted as well.

Response: It has been highlighted in Table 3, we would like to add that this result is based on few individuals.

4.f. Was ordinal logistic regression considered as an analysis? 

Response: Yes, thank you for this comment. Ordinal logistic regression and quantile regression was considered. However, the assumptions for ordinal logistic regression were not met and therefore we decided to treat the outcome as a nominal variable. The resulting model fit was good and the results were in line with our anticipated clinical interpretation.

  1. Are the conclusions consistent with the evidence and arguments
    presented and do they address the main question posed?

Overall yes, however:

lines 228-9, 248-9: novel finding may not be the best term here. The findings were null or in unexpected directions. As is typical in these situations, it's not at all clear why this was the case. Perhaps the authors are correct that there were cultural/social structural factors that have mitigated or reversed some associations, but it is not knowable here.  

Response: Thank you for this comment. We agree that some of our findings are difficult to interpret and the direction of some of the revealed association is unexpected. A possible explanation might be the presence of hidden confounders as for example cultural/social structural factors, which might have impacted the association. Therefore, we have modified the wording as follows. Line: 235-241.

The lack of association between maternal perceived stress during pregnancy or post-partum and iSC levels is an unexpected finding and has to our knowledge previously not been seen in other studies. Infant cortisol stress reactivity score has previously been associated with maternal stress (16) and maternal distress in pregnancy (27). In contrast to our results, Leung et al. (16) found no association between iSC and feeding, health indicators, or sociodemographic. We did not measure stress reactivity in the infant in this study. This could be a possible explanation why our results differ.

Previous text: The lack of association between maternal perceived stress during pregnancy or post-partum and infant SC levels is a novel finding and has to our knowledge previously not been published. Infant cortisol stress reactivity score has previously been associated with maternal stress (23) and maternal distress in pregnancy (24). We did not measure stress reactivity in the infant in this study. This could be a possible explanation why our results differ.

  1. Are the references appropriate?

Yes.

  1. Please include any additional comments on the tables and figures.

Tables in general need better formatting for clarity- they're difficult to read across multiple pages. Sometimes the headings contain notes that should actually be elaborated in a formal table footnote for clarity. In other cases a table note should better explain what is being presented. For ex., Supplemental Table 3 presents sig. values for pairwise test against a reference catgeory within each factor. Suppl. 4 presents tests within boy and girl strata.

Response: Thank you for this comment. We have added description to Supplementary Figure 1. The description was missing in the manuscript format .doc and .pdf sent to Children. It was however included in the .pdf with figures. We apologize for this shortcoming.

Supplementary Figure 1. Infant saliva cortisol median levels related to sampling time (n=1057) over 24 hours. Values are given in median (blue large squares) and min-max (grey small squares). MT=missing time of saliva sampling (n= 427).

We have edited Supplementary Table 1-4 to clarify the content. We have also tried to compress the tables for readability. Supplementary Table 2 was unfortunately unable to fit into one page alone in this upright format.

Round 2

Reviewer 1 Report

It is clear the analysis has many limitations because the over-arching study was not designed to answer specific questions regarding maternal stress and cortisol levels in mothers and infants. I would still like to see an expanded limitations section in the discussion with more focuses on limitations, rather than strengths. I would also like to see the the term "exploratory study" used consistently throughout the manuscript from the abstract through to the conclusion.

Author Response

  1. It is clear the analysis has many limitations because the over-arching study was not designed to answer specific questions regarding maternal stress and cortisol levels in mothers and infants. I would still like to see an expanded limitations section in the discussion with more focuses on limitations, rather than strengths.

We have revised the section in the discussion regarding strength and limitations as follows, to further expand on the limitations.

Strengths and limitations, Line: 325-330

The following limitations should be noted. The population in this retrospective exploratory study originated from the PreventADALL birth cohort, were the main objective focuses on allergic disease prevention and the development of non-communicable diseases. The proportion of participants with a heredity of allergic diseases could therefore be biased. Furthermore, we used one validated instrument to measure stress in the mothers. That makes the generalizability of this study limited. The sampling method used resulted in a large amount of missing saliva samples (1057/2131) and missing records of…

Previous text:

Limitations in this study was the sampling method which resulted in a large amount of missing saliva samples (1057/2131) and missing records of sampling time of the infant saliva due to the home-sampling procedure.

We have added, Line: 335-338

We did not collect complementary information nor further investigate if instructions to study protocol was followed. Our stress data was based on one validated instrument and it was collected through self-reported electronic questionnaires which could increase the risk of information bias.

We have removed:

However, our findings will add to the knowledge of iSC in the general population without a predefined, well controlled clinical setting.

  1. I would also like to see the term "exploratory study" used consistently throughout the manuscript from the abstract through to the conclusion.

Response: To clarify, we have added the term “exploratory study” at a number of places in the manuscript, for detailed information see below. We believe, if exploratory was added throughout the manuscript where study is written, this could interfere with readability. With these changes, we believe that the message is clear and that the readability is not affected.

Abstract, Line 34:

Results: In this exploratory study neither PSS at any time… 

Previous text:

Results: Neither PSS at any…

Aim, Line 72: The aims of this exploratory study were to analyze the associations between maternal…

Previous text:

The aims of this study were to explore the relationship between maternal…

Materials and methods, Line 83:

The present exploratory study adheres…

Previous text:

The present study adheres…

Discussion, Line 225:

In this exploratory study with 1057 infants from the general population….

Previous text:

In 1057 infants from a general population-based mother child birth-cohort..

Conclusions, Line 355:

In this exploratory study, maternal perceived stress in pregnancy and post-partum was not associated with iSC levels at three-months age.

Previous text:

Maternal perceived stress in pregnancy and post-partum was not associated with iSC levels at three-months age.